# Baseline Splenic Volume as a Prognostic Biomarker of FOLFIRI Efficacy and a Surrogate Marker of MDSC Accumulation in Metastatic Colorectal Carcinoma

**DOI:** 10.3390/cancers12061429

**Published:** 2020-05-31

**Authors:** Julie Niogret, Emeric Limagne, Marion Thibaudin, Julie Blanc, Aurelie Bertaut, Karine Le Malicot, Yves Rinaldi, François-Xavier Caroli-Bosc, Franck Audemar, Suzanne Nguyen, Corinne Sarda, Catherine Lombard-Bohas, Christophe Locher, Miguel Carreiro, Jean-Louis Legoux, Pierre-Luc Etienne, Mathieu Baconnier, Marc Porneuf, Thomas Aparicio, Francois Ghiringhelli

**Affiliations:** 1Department of Medical Oncology, Georges François Leclerc Cancer Center-UNICANCER, 1 rue Professeur Marion, 21000 Dijon, France; jniogret@cgfl.fr; 2Department of Medical Oncology, University of Burgundy-Franche-Comté, 7 Boulevard Jeanne d’Arc, 21000 Dijon, France; karine.le-malicot@u-bourgogne.fr; 3INSERM U1231, 7 Boulevard Jeanne d’Arc, 21000 Dijon, France; 4Platform of Transfer in Cancer Biology, Georges François Leclerc Cancer Center—UNICANCER, 1 rue Professeur Marion, 21000 Dijon, France; elimagne@cgfl.fr (E.L.); mthibaudin@cgfl.fr (M.T.); 5Methodology, Data-Management, and Biostatistics Unit, Georges François Leclerc Cancer Center—UNICANCER, 1 rue Professeur Marion, 21000 Dijon, France; jblanc@cgfl.fr (J.B.); abertaut@cgfl.fr (A.B.); 6Fédération Francophone de Cancérologie Digestive, EPICAD INSERM U1231, 7 Boulevard Jeanne d’Arc, 21000 Dijon, France; 7Department of Hepato-Gastroenterology, European Hospital, 6 Rue Désirée Clary, 13003 Marseille, France; yrinaldi@wanadoo.fr; 8Department of Gastroenterology, University Hospital Center, 4 Rue Larrey, 49100 Angers, France; FXCaroli-Bosc@chu-angers.fr; 9Department of Gastroenterology, Côte Basque Hospital Center, 13 Avenue de l’Interne Jacques Loeb, 64100 Bayonne, France; faudemar@ch-cotebasque.fr; 10Department of Medical Oncology, Hospital Center, 4 Boulevard Hauterive, 64000 Pau, France; suzanne.nguyen@ch-pau.fr; 11Department of Medical Oncology, Saintonge Hospital Center, 11 Boulevard Ambroise Paré, 17100 Saintes, France; sarda.co@orange.fr; 12Department of Medical Oncology, Edouard Herriot Hospital, HCL, 5 Place d’Arsonval, 69003 Lyon, France; catherine.lombard@chu-lyon.fr; 13Department of Gastroenterology, Est-Francilien Great Hospital, 6-8 Rue Saint-Fiacre, 77100 Meaux, France; clocher@ghef.fr; 14Department of Medical Oncology and Internal medicine, Hospital Center, 100 Rue Léon Cladel, 82000 Montauban, France; m.carreiro@ch-montauban.fr; 15Department of Hepato-Gastroenterology and Digestive Oncology, Regional Hospital Center, 14 Avenue de l’Hôpital, 45100 Orléans, France; jean-louis.legoux@chr-orleans.fr; 16Department of Medical Oncology, CARIO, Côtes d’Armor Private Hospital, 10 Rue François Jacob, 22190 Plerin, France; pl.etienne@wanadoo.fr; 17Department of Hepato-Gastroenterology, Annecy Genevois Hospital Center, 1 Avenue de l’Hôpital, 74374 Pringy, France; mbaconnier@ch-annecygenevois.fr; 18Department of Medical Oncology and Hematology, Yves Le Foll Hospital Center, 10 Rue Marcel Proust, 22000 Saint-Brieuc, France; marc.porneuf@ch-stbrieuc.fr; 19Department of Gastroenterology, University Hospital Center Saint Louis, APHP, 1 Avenue Claude Vellefaux, 75010 Paris, France; thomas.aparicio@aphp.fr

**Keywords:** metastatic colorectal cancer, splenomegaly, circulating monocytic myeloid derived suppressor cells, MDSC, prognostic biomarker

## Abstract

Background: Predictive biomarkers of response to chemotherapy plus antiangiogenic for metastatic colorectal cancer (mCRC) are lacking. The objective of this study was to test the prognostic role of splenomegaly on baseline CT scan. Methods: This study is a sub-study of PRODIGE-9 study, which included 488 mCRC patients treated by 5-fluorouracil, leucovorin and irinotecan (FOLFIRI) and bevacizumab in first line. The association between splenic volume, and PFS and OS was evaluated by univariate and multivariable Cox analyses. The relation between circulating monocytic Myeloid derived suppressor cells (mMDSC) and splenomegaly was also determined. Results: Baseline splenic volume > 180 mL was associated with poor PFS (median PFS = 9.2 versus 11.1 months; log-rank *p* = 0.0125), but was not statistically associated with OS (median OS = 22.6 versus 28.5 months; log-rank *p* = 0.1643). The increase in splenic volume at 3 months had no impact on PFS (HR 0.928; log-rank *p* = 0.56) or on OS (HR 0.843; log-rank *p* = 0.21). Baseline splenic volume was positively correlated with the level of baseline circulating mMDSC (*r* = 0.48, *p*-value = 0.031). Conclusion: Baseline splenomegaly is a prognostic biomarker in patients with mCRC treated with FOLFIRI and bevacizumab, and a surrogate marker of MDSC accumulation.

## 1. Introduction

Colorectal cancer (CRC) is the second most commonly diagnosed cancer in Europe and a leading cause of death both in Europe and worldwide [1,2]. In 2012, there were 447,000 new cases of CRC in Europe, with 215,000 deaths, and worldwide there were 1.4 million new cases, with 694,000 deaths. About 20–25% of patients with CRC present with metastatic disease at the time of diagnosis, and a further 20–25% of patients will develop metastases later [3]. For metastatic CRC (mCRC), the typical chemotherapy backbone comprises a fluoropyrimidine (intravenous 5-FU or oral capecitabine) used in various combinations and schedules with irinotecan or oxaliplatin [4]. The VEGF antibody (bevacizumab) and EGFR antibodies (cetuximab and panitumumab) are frequently used in combination with chemotherapy [5]. Currently, the classical first line chemotherapies consist in bi-chemotherapy, comprising an injection of irinotecan, fluorouracil and leucovorin (FOLFIRI) or the combination of an injection of oxaliplatin, fluorouracil and leucovorin (FOLFOX) with a biotherapy targeting VEGF or EGFR. More recently, tri-chemotherapy has come to be used in first line for aggressive disease [6,7]. The concept of treatment discontinuation has also recently been introduced, and after “induction” treatment, an active maintenance treatment is seen as a possible option [8,9]. Today, the median overall survival for patients with mCRC is about 30 months [5]. 

In mCRC, predictive and prognostic biomarkers are used to guide the therapeutic strategy. *RAS* mutational status predicts the absence of efficacy of EGFR antibody therapies in mCRC [10,11]. *BRAF* mutations are a significant negative prognostic marker [12], and are predictive of the efficacy of combined EGFR, BRAF and MEK tyrosine kinase inhibition [13,14]. Microsatellite instability (MSI) is another negative prognostic marker in the metastatic setting [15] and a strong predictive marker of the efficacy of immune checkpoint inhibitors [16]. Dihydropyrimidine dehydrogenase (DPD) activity has been shown to predict potential toxicity when using 5-FU and capecitabine [17], while UDP glucuronosyltransferase 1 family polypeptide A1 (UGT1A1) gene polymorphisms are predictive of irinotecan-related side-effects (diarrhea, neutropenia and vomiting) [18,19]. Tumor sidedness and consensus molecular classification could also be used to predict prognostic and response to biochemotherapies [20,21,22,23,24]. All these biomarkers are used, in combination with clinical markers such as the patient’s performance status, tumor burden and comorbidity, to stratify patients and determine the optimal therapeutic strategy.

Currently, biomarkers to predict response to the combination chemotherapy FOLFIRI plus bevacizumab in mCRC and help with clinical decision-making are needed. The factors associate with shorter overall survival (OS) in PRODIGE 9 trial were WHO performance status > 2, unresected primary tumor, age over 65 were and *BRAF* mutant tumor [25]. Another analysis revealed that high baseline leukocytes count and the lack of carcino-embryonic antigen (CEA) decrease level at first evaluation were associated with early progression [26]. Moreover, a radiomic signature at baseline and 2-month CT was able to predict OS [27]. Myeloid-derived suppressor cells (MDSC) can support tumor progression and have been shown to accumulate in the blood and peripheral lymphoid organs, such as the spleen, in animal models of cancer, leading to splenomegaly [28]. We and others described an accumulation of circulating MDSC in patients with mCRC [29,30,31] and in those with pancreatic cancer [32,33,34] as compared to healthy donors. A high level of circulating MDSC at baseline is significantly associated with poor progression-free survival (PFS) and poor OS in mCRC and pancreatic cancer [29,33,34,35]. In addition, our group [36] observed that baseline splenomegaly is a predictive marker of poor response to FOLFIRINOX in advanced pancreatic carcinoma. Together, these data suggest that MDSC level could be a surrogate marker associated with better PFS. The assessment of circulating MDSC levels is not routinely performed, but splenomegaly could be a surrogate marker of MDSC levels.

In this prospective cohort study based on the PRODIGE 9 population, we first aimed to determine the prognostic role of baseline splenomegaly and chemotherapy-induced splenomegaly in mCRC patients treated with first line FOLFIRI. Secondly, we aimed to determine whether splenic volume is correlated with the rate of circulating MDSC.

## 2. Results

### 2.1. Population Based Prospective Cohort

Out of the 488 patients included in the modified intention-to-treat population of the PRODIGE-9 study, 266 with available CT scan and written informed consent for sub-studies were eligible for the present analysis. Of these, 14 were excluded because their CT scan was not amenable to measurement of the spleen because of technical problems or splenectomy (Figure 1). Two hundred and fifty-two patients were thus included in the present analysis. The characteristics of the cohort were representative of those of the PRODIGE-9 patients (Table 1). Only 55 patients (21.8%) had received adjuvant chemotherapy for localized CRC before inclusion in PRODIGE-9.

### 2.2. Association between Baseline Splenic Volume and Progression-Free Survival

Overall, at the last follow-up, 248 patients progressed or died; only 4 patients did not progress and were censored. We first analyzed baseline splenic volume as a continuous variable. Univariate Cox analyses identified nine factors significantly (*p* < 0.05) associated with PFS (Table 2), including baseline splenic volume (HR 1.001; 95CI% [1–1.002]; log-rank *p* = 0.05). Multivariate Cox analyses identified four significant prognostic factors for PFS (Table 2), including baseline splenic volume (HR 1.001; 95%CI [1.000–1.0003]; log-rank *p* = 0.01). Next, we analyzed baseline splenic volume as a binary variable. Using the median of baseline splenic volume (214 mL) as the cut-off, no difference in PFS was demonstrated. We then sought to identify the threshold with the best discriminatory power, using Cutoff Finder software. The baseline splenic volume threshold retained was 180 mL with a sensitivity of 67.3% (61.3–72.9%) and a specificity of 50% (15–85%). Using this threshold in univariate Cox analysis, baseline splenic volume was identified as a prognostic factor for PFS (HR 1.403; 95%CI [1.073–1.834]; log-rank *p* = 0.01). By multivariate Cox analysis, the baseline splenic volume threshold of 180 mL was found to be significantly associated with PFS (HR 1.362; 95%CI bootstrapped [1.338–1.370]; *p* = 0.0249) (Table 3), and the stability of the model was verified using a bootstrap with 500 replications. Harrell’s C-index for the baseline splenic volume threshold at 180 mL was 0.84 (95%CI [0.75–0.93]). In our cohort, baseline splenic volume was ≤180 mL in 83 patients (32.9%), and > 180 mL in 169 patients (67.1%). The Kaplan–Meier curves showed that baseline splenic volume > 180 mL was associated with poor PFS (median PFS = 9.2 vs. 11.1 months, >180 mL vs. ≤180 mL respectively; log-rank *p* = 0.0125) (Figure 2A).

### 2.3. Association Between Baseline Splenic Volume and Overall Survival

In our cohort, 216 patients died. We analyzed baseline splenic volume as a binary variable, and tested the previously identified threshold of 180 mL. Using this threshold, baseline splenic volume was not significantly associated with OS by univariate Cox analyses (HR 1.223; 95%CI [0.920–1.624]; log-rank *p* = 0.1654) or by multivariate Cox analyses (HR 1.094; 95%CI [0.819–1.461]; log-rank *p* = 0.5419) (Appendix A). Baseline splenic volume > 180 mL was not statistically significantly associated with OS (median OS = 22.6 vs. 28.5 months, >180 mL vs. ≤180 mL respectively; log-rank *p* = 0.1643) (Figure 2B). 

Association between the splenic volume at 3 months and patient’s prognosis in our cohort, after 3 months of treatment, we observed a decrease in splenic volume in 110 patients (43.8%), and an increase in 141 patients (56.2%). The increase in the splenic volume at 3 months was not associated with PFS (HR 0.928; log-rank *p* = 0.56) nor OS (HR 0.843; log-rank *p* = 0.21).

### 2.4. Association Between Baseline Splenic Volume and Circulating MDSC

In a cohort of 19 still untreated mCRC patients from another cohort (clinical trial MEDITREME), we evaluated splenic volume at baseline, as well as the levels of various blood immune cell populations. We used 10 age- and sex- matched healthy volunteers as controls. We observed that patients with mCRC had baseline higher myeloid cell counts, and lower lymphoid cell counts than healthy controls (Figure 3A,B). Regarding the myeloid subset, mCRC patients had significantly higher levels of granulocytes, monocytic MDSC (mMDSC) and lower levels of activated monocytes than healthy donors (Figure 3C). Baseline splenic volume was positively correlated with the level of baseline circulating neutrophils and mMDSC (Figure 3D,E). Moreover, baseline splenic volume was inversely correlated with the number of lymphocytes and activated monocytes (Figure 3F,G), but not with other immune parameters, thus suggesting that splenic volume is a surrogate marker of an unfavourable peripheral immune response with accumulation of immunosuppressive myeloid cells.

## 3. Discussion

This study underlines that splenic volume could be a predictive marker of PFS in patients treated with FOLFIRI plus bevacizumab. As splenic volume appears to be a surrogate marker of mMDSC levels, this study raises the hypothesis that mMDSC could modulate the efficacy of FOLFIRI plus bevacizumab in mCRC.

In line with other studies [3,37,38], we observed that Köhne criteria, number of metastatic sites, resection of primary tumor, and baseline level of alcaline phosphatase, platelets, leukocytes, and LDH were prognostic factors for survival. Previous ancillary studies in PRODIGE9 study confirmed the poor prognostic role of high leukocytes count and show that lack of CEA decrease is also an important factor of poor prognosis [26]. Imaging-based predictive factor was also tested in PRODIGE 9 cohort and a radiomic signature at baseline and 2-month CT was able to predict OS and identify good responders better than RECIST1.1 [27]. However, our study is the first to observed that splenic volume is associated with PFS in mCRC, and it confirms our previous findings in patients treated with FOLFIRINOX for advanced pancreatic cancer [36]. Our work suggests that baseline splenic volume, used as either a continuous or binary variable, could be a new prognostic marker of FOLFIRI plus bevacizumab efficacy in first-line treatment for mCRC, insofar as we show that baseline splenic volume >180 mL is associated with poor PFS. Additional studies are required to validate this cut-off, and also to determine whether it is a new prognostic marker or predictor of the efficacy of FOLFIRI plus bevacizumab. We expect that new studies in patients treated with anti-EGFR or FOLFOX-based chemotherapy will improve the comprehension of our results. Since normal splenic volume ranges between 110 and 340 mL [39], we cannot rule out the possibility that a splenic volume threshold of 180 mL as selected in this study may be suboptimal, and this warrants confirmation in another study. In addition, the cut off determine is this study differ from our study in pancreatic cancer which select a cut off of 340 mL. We believe that such data could be explained by an anatomic and a biological hypothesis. First, in pancreatic cancer, many patients have portal hypertension which could lead to splenic enlargement. Secondly neutrophilia and accumulation of MDSC is more frequent in pancreatic disease than colorectal cancer. 

Abnormal differentiation and function of myeloid cells is a hallmark of cancer. MDSC can support tumor progression by promoting tumor cell survival, angiogenesis, invasion of healthy tissue by tumor cells, and metastases [40]. MDSC are a heterogeneous population of myeloid cells with either a granulocyte or monocytic phenotype and characterized by their capacity to exert immunosuppressive functions [41]. They are also suspected of exerting their deleterious effect via their capacity to blunt T dependent antitumor immune response [42]. In tumor-bearing hosts, MDSC accumulate in peripheral lymphoid organs, such as the spleen and tumor tissues [43]. Marked splenomegaly related to MDSC accumulation is a classical observation in many experimental models of transplantable cancer in MDSCs. In humans, MDSC correlates with clinical outcomes and is an independent prognostic indicator of clinical disease progression in patients with pancreatic cancer, esophageal cancer, gastric cancer, and melanoma [34,43]. In mCRC, we previously demonstrated that there is an accumulation of circulating MDSC compared to healthy donors, and high levels of circulating MDSC at baseline are associated with poor PFS and OS [29]. Moreover, in mCRC, treatment with FOLFOX plus bevacizumab induces a decrease in MDSC, which in turn is associated with better PFS [29]. Likewise, Tada et al. showed that mCRC patients with high mMDSC and low blood T cell levels had significantly shorter PFS [35]. Our results are consistent with previous studies, as we observed a higher rate of circulating MDSC in patients with mCRC than in healthy donors. 

The circulating MDSC level could represent an early marker of disease progression [44], but analysis of MDSC levels is complex and not yet possible in routine. In animal models, MDSC accumulation is associated with splenomegaly [28]. To the best of our knowledge, this is the first time that a positive correlation has been demonstrated between splenic volume and circulating MSDC levels in humans. Thus, we believe that baseline splenic volume could be used as a surrogate marker of MDSC accumulation. As MDSC level is associated with poor prognosis in mCRC, cancer therapy targeting this immunosuppressive cell type could be of therapeutic interest. Inhibitors of CXCR4, CCL2 and VEGF are known to affect MDSC levels and are currently in development to combat MDSC dependent immunosuppression [45]. Splenomegaly may be a surrogate marker of the efficacy of such therapies.

Our study has several strengths. We evaluated a large, multicenter cohort of patients with mCRC generated from a multicenter phase III trial. Thus, clinical patient characteristics were homogeneous and limit potential for bias. In addition, bootstrapping enabled us to assign measures of accuracy and internal validation of the prognostic role of spleen volume (defined in terms of bias, variance, confidence intervals, and prediction error). However, a validation cohort in another prospective cohort of patients treated with FOLFIRI bevacizumab before using spleen volume as a prognostic marker in mCRC. In addition, further studies with different treatment strategies are warranted to determine the prognostic versus predictive role of this new marker.

## 4. Methods and Materials

### 4.1. Patient Selection and Study Design

PRODIGE 9 was a cooperative, multicenter, open-label, phase III randomized controlled trial, conducted by the Fédération Francophone de Cancérologie Digestive and the PRODIGE (Partenariat de Recherche en Oncologie DIGEstive) group in 66 French centers [25,46]. From March 2010 to July 2013, 491 patients were included. Main eligible criteria were histologically proven, unresectable mCRC, WHO status ≤ 2, life-expectancy ≥ 3 months, and absence of previous chemotherapy or antiangiogenic therapy for metastatic disease [25]. All patients provided written informed consent, and the study was approved by the Committee for the Protection of Persons Ile de France VIII. The trial was registered on ClinicalTrials.gov (NCT00952029). All patients were treated with FOLFIRI and bevacizumab for 6 months. After this induction treatment the patients were randomized in a bevacizumab maintenance versus no treatment (observation) arm during the chemotherapy free interval (CFI). After disease progression during the maintenance or pause, the induction regimen was repeated for eight cycles, followed by a new CFI. In the present cohort, we included all PRODIGE 9-patients for whom computed tomography (CT) scan was available for central review at baseline, 3 months and 6 months. Only patients with available CT scan and who provided written informed consent for substudies were included in the present analysis. Patients with splenectomy were excluded. 

To investigate the correlation between MDSC level and splenomegaly, we used a subset of patients include in another first line clinical trial MEDITREME (NCT03202758). Blood from age- and sex-matched healthy donors was obtained from Fench Blood Transfusion Service.

### 4.2. Spleen Volume Estimation

Spleen volume was measured by CT scan as previously described [46]. The maximal width (W) of the spleen, determined as the largest diameter on any transverse section, the thickness at the hilum (Th), determined as the distance between the inner and outer borders of the spleen on a plane perpendicular to the splenic width and through the hilum, and length (L) were obtained from abdominal CT examinations. Spleen volume was calculated using the formula: Spleen volume = 30 + 0.58 (W × L × Th.). A value between 110 and 340 mL is considered normal [46]. Spleen measurements and volume calculations were performed by the same investigator, who was blinded to clinical outcomes. Splenomegaly was defined as spleen volume > 340 mL.

### 4.3. Flow Cytometry

Myeloid subpopulations were identified using custom Duraclone tubes (Beckman Coulter), using anti-CD15-Pacific Blue, anti-CD33-FITC, anti-CD3-PC5.5, anti-CD19-PC5.5, anti-CD20-PC5.5, anti-CD56-PC5.5, anti-HLA-DR-AA750 and anti-CD14-Krome Orange dry antibodies (all obtained from Beckman Coulter). We added in liquid form anti-CD11b-BV605 antibody (Biolegend) and Draq7 viability dye (Beckman Coulter). Whole blood was removed to heparinized tubes (100 µL) and stained in Duraclone tubes for 15 min at room temperature. After surface staining, 2 mL of Versa Lyse/IOTest 3 solution (Beckman Coulter) was added for 10 min. Cells were then centrifuged at 350g for 5 min). All events were acquired on a BD CANTO2 cytometer equipped with BD FACSDiva software (BD Pharmingen, Le Pont De Claix, France), and data were analyzed using Kaluza software (Beckman Coulter). The gating strategy used to analyze myeloid cell subsets has previously been described elsewhere [29]. Briefly, singlet live blood CD11b^+^ Lineage^−^ leukocytes were considered as myeloid cells and CD11b^−^ Lineage^+^ as total lymphocytes. Then, we identified granulocytes as CD15^+^ CD33^−^, granulocytic MDSC (gMDSC) as CD15^+^ CD33^+^, monocytic MDSC (mMDSC) as CD15^−^ CD33^+^ CD14^+^ HLA-DR^low/neg^ and mature monocytes as CD15^−^ CD33^+^ CD14^+^ HLA-DR^high^. For HLA-DR positivity identification, we used FMO controls. The gating strategy is presented in the Appendix A.

### 4.4. Outcomes

The primary objective was to determine whether baseline splenic volume is an independent prognostic factor of PFS in patients with mCRC treated by FOLFIRI plus bevacizumab, and if so, to determine the cut-off of splenic volume associated with PFS.

Secondary objectives were to determine whether baseline splenic volume is a prognostic factor of OS in patients with mCRC treated by FOLFIRI plus bevacizumab in first line; whether splenic volume size modification at 3 months was associated with PFS or OS, and to examine the correlation between splenic volume and circulating MDSC levels.

### 4.5. Statistical Analysis

Categorical variables are described as number and percentage, excluding missing data. Continuous variables are described as median and range. Median follow-up was calculated according to the reverse Kaplan–Meier method. The response to treatment was determined on CT scan using RECIST version 1.1 criteria. PFS was defined as the time between randomization and first progression or death from any cause. OS was defined as the time between randomization and death from any cause. Patients without event were censored at the date of last follow-up. Survival curves and median survival (with 95% confidence interval) were estimated using the Kaplan–Meier method. The log rank test was used to compare survival curves. Baseline splenic volume was analyzed as a continuous variable and its prognostic role on OS and PFS was determined by univariate and multivariate Cox models. Variables with a *p*-value < 0.10 by univariate analysis were entered into the multivariate model. The multivariate model was adjusted for spleen volume at baseline. The correlation between variables included in the model was tested, and if the correlation coefficient between two variables was >0.5, only the more clinically relevant of the two, or the variable with the most significant Hazard Ratio (HR) was entered into the model. In case of non-compliance with the log linearity condition of the variable “spleen volume”, a logarithmic transformation could be used. A bootstrap approach with 500 replications was used to check for the stability of the model. The predictive power of the model was assessed using Harrell’s C index.

Analyses were repeated including the variable “splenic volume” as a binary variable. The threshold was determined using the best cut-off method using Cutoff Finder software (Charité-Universitätsmedizin Berlin, Berlin Germany) [47] and analyzed in a multivariate model.

Correlations between blood immune parameters and spleen volume were performed using the non-parametric Spearman correlation test.

Tests were two-sided and a *p*-value < 0.05 was considered significant. All analyses were performed using SAS software version 9.4 (SAS Institute Inc., Cary, NC, USA).

## 5. Conclusions

In conclusion, we propose that baseline splenomegaly could be a prognostic marker of FOLFIRI plus bevacizumab efficacy in mCRC, and a surrogate marker of MDSC accumulation. These results warrant confirmation in other studies with FOLFIRI plus bevacizumab treatment, and of course, in studies using other drugs.

## Figures and Tables

**Figure 1 cancers-12-01429-f001:**
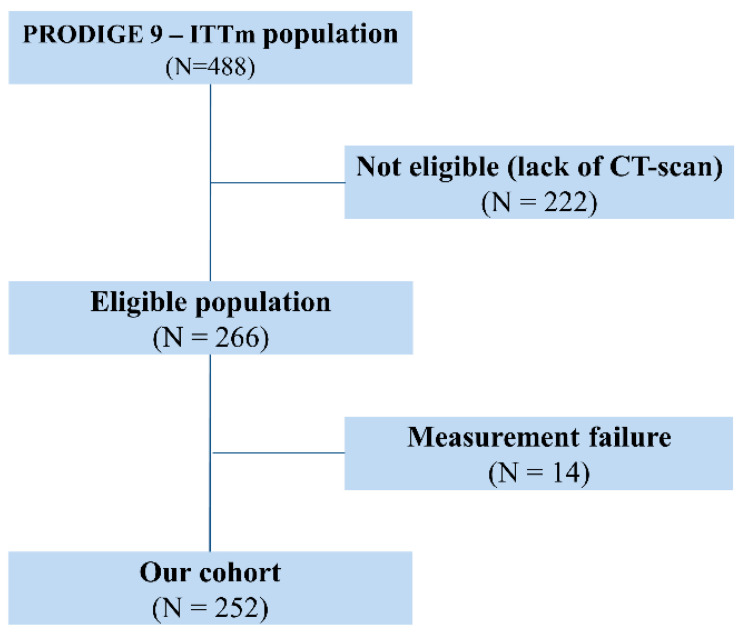
Flow chart of the study population.

**Figure 2 cancers-12-01429-f002:**
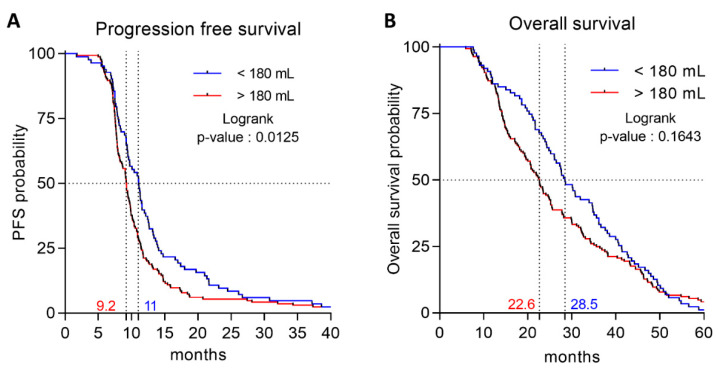
Association between spleen volume and progression-free (PFS) and overall survival (OS). (**A**) Kaplan–Meier survival curves for PFS according to spleen volume (**B**) Kaplan–Meier survival curves for OS according to spleen volume.

**Figure 3 cancers-12-01429-f003:**
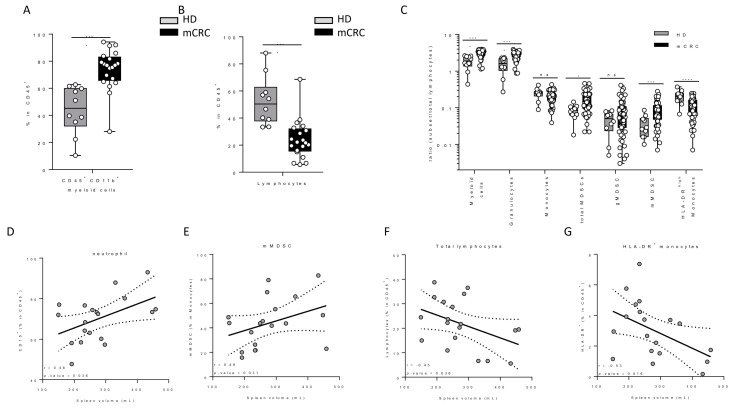
Association between blood parameters and spleen volume. (**A**,**B**) Comparison of myeloid and lymphocyte cell proportions in mCRC and peripheral blood leukocytes from matched healthy donors (HD). (**C**) Myeloid subset/lymphocyte ratio in mCRC and matched healthy donors. (**D**–**G**) Correlation between spleen volume and the proportion of granulocytes, mMDSC, lymphocytes and activated HLA-DR^+^ monocytes respectively.

**Table 1 cancers-12-01429-t001:** Baseline characteristics of the modified intention to treat population of the overall PRODIGE 9 trial, and of the cohort included in the present analysis.

Characteristic	Our Cohort (*n* = 252)	Prodige 9 (*n* = 488)
Median age, years	64.3	64.2
Male gender	177 (70.2)	316 (64.8)
Primary tumor resected,	148 (58.7)	281 (57.6)
WHO performance status		
0	132 (52.4)	233 (47.7)
1	107 (42.5)	216 (44.3)
2	13 (5.2)	39 (8%)
Köhne criteria		
Low	87 (34.5)	181 (37.1)
Intermédiate	126 (50)	216 (44.3)
High	39 (15.5)	91 (18.6)
Number of metastatic sites		
1	91 (36.1)	192 (39.3)
>1	161 (63.9)	296 (60.7)
Location		
Right colon	55 (30.1)	100 (28.1)
Left colon	71 (38.8)	157 (44.1)
Rectum	57 (31.1)	99 (27.8)
Missing	69	132
Mutated KRAS	82 (41)	173 (46.1)
Missing	52	113
Mutated BRAF	9 (6.5)	21 (8.6)
Missing	114	243
Randomization arm		
Maintenance	123 (48.8)	245 (50.2)
Observation	129 (51.2)	243 (49.8)

**Table 2 cancers-12-01429-t002:** Factors associated with Progression-Free Survival using baseline splenic volume as a continuous variable, in univariate and multivariate Cox analyses. *HR, hazard ratio; CI, confidence interval, LDH, Lactate deshydrogenase.*

Variable	HR	95%CI	*p*-Value	HR	95%CI	*p*-Value
Baseline splenic volume	1.001	[1.000–1.002]	0.05	1.001	[1.000–1.0003]	0.01
Köhne criteria						
High vs. Low	1.672	[1.143–2.447]	0.03			
Intermediate vs. Low	1.147	[0.870–1.513]				
Primary tumor resected, yes vs no	0.717	[0.556–0.926]	0.01	0.731	[0.564–0.947]	0.02
Number of metastatic sites, >2 vs. ≤2	1.468	[1.092–1.972]	0.01	1.512	[1.124–2.034]	0.01
Baseline Alcaline Phosphatase, >300 U/L vs. ≤300 U/L	1.536	[1.099–2.147]	0.01			
Baseline Leukocytes, >10 G/L vs. ≤10 G/L	1.491	[1.104–2.012]	0.01	1.487	[1.093–2.022]	0.01
Baseline Platelets, >300 G/L vs. ≤300 G/L	1.321	[1.030–1.696]	0.03			
Baseline Neutrophils, >5.2 G/l vs. ≤5.2 G/L	1.277	[0.994–1.641]	0.06			
Baseline LDH, >340 U/L vs. ≤340 U/L	1.353	[1.035–1.770]	0.03			

**Table 3 cancers-12-01429-t003:** Factors associated with Progression-Free Survival using baseline splenic volume as a vinary variable, in multivariate Cox analysis with bootstrapping. *HR, hazard ratio; CI, confidence interval.*

Factors	HR	95%CI	95%CI Bootstrapped	*p*-Value
Baseline splenic volume >180mL vs. ≤180 mL	1.362	[1.040–1.784]	[1.338–1.370]	0.02
Primary tumor resected, yes vs. no	0.719	[0.557–0.929]	[0.709–0.726]	0.01
Number of metastatic sites, >2 vs. ≤2	1.549	[1.150–2.086]	[1.537–1.578]	0.01
Baseline Alcaline Phosphatase >300 U/L vs. ≤300 U/L	1.512	[1.079–2.117]	[1.483–1.526]	0.02

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
