# Peer review of "Baseline Splenic Volume as a Prognostic Biomarker of FOLFIRI Efficacy and a Surrogate Marker of MDSC Accumulation in Metastatic Colorectal Carcinoma"

_cancers, 2020, doi:10.3390/cancers12061429_

Round 1

Reviewer 1 Report

In this substudy analysis of PRODIGE9 patients, the authors evaluated whether splenic volume may function as a predictive biomarker of response to chemotherapy.  More specifically, the group evaluated baseline splenic volume as an independent prognostic factor of PFS in patients treated with FOLFIRI/bevacizumab in the first-line setting.  As secondary objectives, the authors evaluated whether baseline splenic volume is also a prognostic factor of OS, if splenic volume size modification 3 months was also associated with PFS or OS, and investigated correlation between splenic volume and circulating MDSC levels.

The study identified 252 patients out of 488 included in the modified intention to treat population of the PRODIGE9.  The median of baseline splenic volume was 214 ml; however, this was not associated with PFS.  Using Cutoff Finder software the group identified 180 ml as a threshold with the best discriminatory power (sensitivity of 67.3% and a specificity of 50%).  In urivariant analysis, 180 ml baseline splenic volume was noted as a prognostic factor for PFS with HR of 1.403;95%CI[1.073-1.834].  This was also the case when looking at a multi-variant analysis with HR of 1.362;95%CI[1.338-1.70].  However, when the group analyzed association with OS or increase in splenic volume at 3 months, no statistical significance was established.

Finally, in a separate cohort of 19 patients from MEDITREME trial and 10 age- and sex- match healthy volunteers as control. The authors analyzed blood immune cells and found that baseline levels circulating neutrophils and mMDSCs positively correlated with baseline splenic volume.

Overall, this is a very important topic in clinical care.  Identification of prognostic and predictive biomarker will directly impact and guide therapeutic strategies for many of our patients.  The manuscript is well written and clear the science is sound and logical with clear figures and tables.

The primary endpoint reflects a statistically significant association between a baseline splenic volume of 180 ml and PFS, where a baseline splenic volume bigger than 180 ml in this specific cohort was associated with poor PFS.  This is highly important and medically concerning as in the study cohort 67.1% of patients had baseline spleen >180ml, with a median of 210ml. What was the range of baseline splenic volume?  Also, given that the normal baseline spleen size ranges between 110 and 340 ml, the cutoff of 180 ml tend to be small and clinically challenging.  Previous work by the group in pancreatic cancer has noted a bigger cutoff of 340ml.  How dose one consolidate this significant difference in the two populations? And how would the authors invasion this normal size spleen (180ml) to be incorporated clinically as a biomarker.

Previous work analyzing feasibility of biomarkers incorporates validation and control cohorts. The data from the MEDITREME trial does function as a validation cohort, no report baseline splenic volumes. Have the authors considered a validation cohort?  In order to envision splenomegaly as a biomarker it will be imperative to evaluate and validate the current results.

Furthermore, on evaluation of Kaplan-Meier survival curves there seem to be a crossover.  Given that the treatment effect (HR) did not vary over time this may not be considered a minor violation of the primary hypothesis assumption.  In such cases, it will be important to understand for which patients the treatment was worse.  Is there a qualitative interaction driving this crossover at that time?

Finally, the authors evaluated a different cohort of patients from the MEDITREME trial, where they collected 19 blood samples to analyze various myeloid and lymphoid subsets and compared them to 10 age-and sex-matched healthy volunteers. The results show that baseline splenic volume was positively correlated with neutrophils and mMDSC and negatively with lymphocyte and active monocytes. How does this data relate to the primary trial cohort? What was the average size of baseline splenic volume in the MEDITREME trial subset of patients?  What was the baseline splenic volume of the control group?  Figure 3 seem to demonstrate that except for 2 patient all the other patients had a spleen volume larger than 180ml, is that accurate?

Minor points

Subtitle line 164, 167

Author Response

1.Overall, this is a very important topic in clinical care. Identification of prognostic and predictive biomarker will directly impact and guide therapeutic strategies for many of our patients. The manuscript is well written and clear the science is sound and logical with clear figures and tables.

Response : wethanks the reviewer with the comment.

2.  What was the range of baseline splenic volume?

Response : the range is 80-756ml we add this information in the manuscript

We Also, given that the normal baseline spleen size ranges between 110 and 340 ml, the cutoff of 180 ml tend to be small and clinically challenging. Previous work by the group in pancreatic cancer has noted a bigger cutoff of 340ml. How dose one consolidate this significant difference in the two populations? And how would the authors invasion this normal size spleen (180ml) to be incorporated clinically as a biomarker.

Response: We aggree that this diffference is surprising. We believe that such data could be explained by different hypothesis. First, in pancreatic disease many patients have portal hypertension which could lead to splenic enlargement. In addition neutrophilia and accumulation of MDSC is more frequent in pancreatic disease than colorectal cancer. We suggest that splenic volume is a surrogate marker of neutrophil and MDSC accumulation so this  anatomic and biological difference between pancreatic cancer and colorectal cancer make explain this discrepancy.

We add a comment accordingly in the discussion.

3. Previous work analyzing feasibility of biomarkers incorporates validation and control cohorts. The data from the MEDITREME trial does function as a validation cohort, no report baseline splenic volumes. Have the authors considered a validation cohort? In order to envision splenomegaly as a biomarker it will be imperative to evaluate and validate the current results.

Response: currently we could not generate validation cohort. And because of that we perform bootstrap analysis to test the stability of of COX model. In addition we decide to include MEDITREME data. These data are data at baseline CT scan before chemotherapy. Because treatment is different than in PRODIGE 9 and because of the few number of patients we did not test association between splenic volume and response. We aggree that a larger prospective cohort is required to validate spleen volume as a clinically relevant marker. We soften this point in the new discussion

4. Furthermore, on evaluation of Kaplan-Meier survival curves there seem to be a crossover. Given that the treatment effect (HR) did not vary over time this may not be considered a minor violation of the primary hypothesis assumption. In such cases, it will be important to understand for which patients the treatment was worse. Is there a qualitative interaction driving this crossover at that time?

Response: We test interaction between splenic volume and treatment arm in prodige 9, leucocytosis, Kohne criteria and primary tumor resection. We  find no positive interraction. The cross over between curve is very modest is due on less than 5 patients. So we believe that such data are not clinically meanful

5. Finally, the authors evaluated a different cohort of patients from the MEDITREME trial, where they collected 19 blood samples to analyze various myeloid and lymphoid subsets and compared them to 10 age-and sex-matched healthy volunteers. The results show that baseline splenic volume was positively correlated with neutrophils and mMDSC and negatively with lymphocyte and active monocytes. How does this data relate to the primary trial cohort?

Response:We do not have biological sample to test mMDSC level in PRODIGE cohort. So we decide to test prospectively association between MDSC and splenic volume in MEDITREME trial.

6. What was the average size of baseline splenic volume in the MEDITREME trial subset of patients? What was the baseline splenic volume of the control group?

Response:MEDITREME is a single arm study and mesure are taken at baseline. The mediane and mean splenic volume in this study are 253 (149-458) and 271m. These volumes are slighly more important than in PRODIGE 9 but in similar ranges. This difference may rely on inclusion criteria (in MEDITREME study only RAS mutated tumor are included). Because of the small difference between the two group we belive that the two cohort could be compared

7. Figure 3 seem to demonstrate that except for 2 patient all the other patients had a spleen volume larger than 180ml, is that accurate?

Response: Yes the reviewer is correct.

Reviewer 2 Report

In their manuscript entitled “Baseline splenic volume as a prognostic biomarker of FOLFIRI efficacy and a surrogate marker of MDSC accumulation in metastatic colorectal carcinoma” the authors aimed to determine the prognostic role of baseline splenomegaly and chemotherapy-induced splenomegaly in mCRC patients treated with first line FOLFIRI and determine whether splenic volume is correlated with the rate of circulating MDSC.

This article is well written and I found this manuscript interesting and valuable. Results of presented study have potential for use in future clinical trials and could be applied for the prognosis in clinical practice. I have some minor comments for the authors as detailed below.

  1. Table 1 needs a better legend/description. In my opinion, the current description unnecessarily incorporates the results of Table 2. An explanation of the values in brackets (that are percentages) should be included.
  2. Minor language errors and typos.
  3. Figure 3A and B - It should be determined on the figure which part relates to mCRC patients and which to healthy donors (similarly to what was done in Fig. 3C).

Author Response

  1. This article is well written and I found this manuscript interesting and valuable. Results of presented study have potential for use in future clinical trials and could be applied for the prognosis in clinical practice. I have some minor comments for the authors as detailed below

Response : We thanks the reviewer for this comment

2. Table 1 needs a better legend/description. In my opinion, the current description unnecessarily incorporates the results of Table 2. An explanation of the values in brackets (that are percentages) should be included.

Response : We make the requested correction

3. Figure 3A and B - It should be determined on the figure which part relates to mCRC patients and which to healthy donors (similarly to what was done in Fig. 3C).

Response : We make the requested correction

Round 2

Reviewer 1 Report

Appreciate the authors responses.  As mentioned prior this is a very interesting and important research with impact on clinical care.  The responses provides insight and clarification to do the questions provided.

Minor comments

In the results section there are inconsistency with each section title.

Line 72 include a numerical number and un-bolded italic title, similarly in line 90.  Line 119 non-numerical bolded title and finally, lines 127 and 133 are non-numerical, non-bolded non-italic

Similar in the material methods, inconsistencies with the font of title sections

Author Response

we make the requested editing